# Tunicamycin Protects against LPS-Induced Lung Injury

**DOI:** 10.3390/ph15020134

**Published:** 2022-01-24

**Authors:** Khadeja-Tul Kubra, Mohammad A. Uddin, Nektarios Barabutis

**Affiliations:** School of Basic Pharmaceutical and Toxicological Sciences, College of Pharmacy, University of Louisiana Monroe, Monroe, LA 71201, USA; kubrak@warhawks.ulm.edu (K.-T.K.); uddinma@warhawks.ulm.edu (M.A.U.)

**Keywords:** inflammation, acute lung injury, acute respiratory distress syndrome, endothelium, vascular barrier

## Abstract

The pulmonary endothelium is a dynamic semipermeable barrier that orchestrates tissue-fluid homeostasis; regulating physiological and immunological responses. Endothelial abnormalities are caused by inflammatory stimuli interacting with intracellular messengers to remodel cytoskeletal junctions and adhesion proteins. Those phenomena are associated with sepsis, acute lung injury, and acute respiratory distress syndrome. The molecular processes beyond those responses are the main interest of our group. Unfolded protein response (UPR) is a highly conserved molecular pathway resolving protein-folding defects to counteract cellular threats. An emerging body of evidence suggests that UPR is a promising target against lung and cardiovascular disease. In the present study, we reveal that Tunicamycin (TM) (UPR inducer) protects against lipopolysaccharide (LPS)-induced injury. The barrier function of the inflamed endothelium was evaluated in vitro (transendothelial and paracellular permeability); as well as in mice exposed to TM after LPS. Our study demonstrates that TM supports vascular barrier function by modulating actomyosin remodeling. Moreover, it reduces the internalization of vascular endothelial cadherin (VE-cadherin), enhancing endothelial integrity. We suggest that UPR activation may deliver novel therapeutic opportunities in diseases related to endothelial dysregulation.

## 1. Introduction

Our lungs are exposed to environmental pathogens (bacteria, virus) and toxins (cigarette smoke, air pollution, asbestos), counteracted by cytokine-mediated host defense mechanisms. The dysregulation of the endothelial barrier during inflammation results in increased permeability; and accumulation of protein-rich edematous fluid. Lung endothelial hyperpermeability is the hallmark of acute respiratory distress syndrome (ARDS), including the COVID-19-related ARDS [1,2]. Endoplasmic reticulum (ER) stress acts upon those stimuli, to maintain homeostasis and activate unfolded protein response (UPR) [3]. Our recent endeavors aim to support a novel therapeutic possibility against pathologies related to endothelial barrier dysfunction, based on targeted UPR activation.

Barrier functions are strictly regulated by endothelial adherens (AJs) and tight junctions (TJs). During inflammation, leukocyte extravasation is governed by the post-translational modifications of those junctions. Vascular endothelial cadherin (VE-cadherin) is an AJ protein that mediates intracellular integrity through its extracellular domain [4]. Vascular beds have distinct architectures for interendothelial junctions (IEJs); so to replenish functional requirements. VE-cadherin modulates the structural integrity of endothelium- as well as TJs- comprising blood–brain barrier (BBB) [5]. Pro-inflammatory stimuli such as lipopolysaccharide (LPS) and other factors (vascular endothelial growth factors) destabilize VE-cadherin, enhancing internalization and paracellular permeability [6]. The proteolytic processing of ER localized protein is suppressed by LPS. This is an endotoxin derived from the outer layer of Gram-negative bacteria [7].

ER is organized in a tubular network and devises metabolic events of protein synthesis, folding, gluconeogenesis, and lipid synthesis. This secretory pathway (ER) is engaged in protein post-translational modifications; as well as in their targeted delivery to the corresponding intra- or extra-cellular destinations. Nascent polypeptides enter ER lumen through post- and co-translational translocation, where they undergo modifications by molecular chaperones, so to acquire functional 3D conformation [8]. Perturbation of ER homeostasis occurs when the influx of unfolded proteins exceeds ER folding capability, which has been linked to pulmonary fibrosis, lung cancer, and diabetes [3].

The adaptive activities of UPR maintain proteostasis and exert anti-inflammatory activities [9,10,11,12]. That phylogenetically conserved signal transduction pathway is orchestrated by three ER transmembrane proteins: inositol-requiring protein-1α (IRE1α), activating transcription factor-6 (ATF6), and protein kinase RNA (PKR)-like ER kinase (PERK). Those sensors are bound to immunoglobulin heavy chain binding protein/glucose-regulated protein 78 (BiP/Grp78). This molecular chaperon dissociates from those sensors due to its higher affinity for the hydrophobic region of misfolded proteins, activating downstream UPR signaling [13].

Tunicamycin (TM) is a naturally occurring antibiotic-and ER stress inducer- which reduces the glycosylation of asparagine-linked (N-linked) glycoproteins, thus triggering UPR [14]. TM exerts its antibacterial activity by inhibiting UDP-HexNAc: polyprenol-P HexNAc-1-P transferase, which belongs to a family of enzymes catalyzing the biosynthesis of the bacterial cell wall [15]. This nucleoside antibiotic (TM) can also potentiate the antibacterial activity of oxacillin, a semisynthetic penicillinase-resistant, and acid-stable penicillin. Hayakawa et al. [16] reported that P53 inducer TM [17] counteracts the tumor necrosis factor alpha (TNF-α)-induced activation of nuclear factor kappa B (NF-kB). Moreover, TM attenuates the induction of monocyte chemoattractant protein-1 (MCP-1)—which is a pro-inflammatory chemokine— in rat mesangial cells and murine podocytes.

We have shown that ER stress inducer brefeldin A [18] protects against LPS-induced lung endothelial barrier disruption [1]. Moreover, Kifunensine—a UPR suppressor—compromises barrier function [19], GHRH antagonists utilize UPR to exert their protective activities in the lungs [20], and Hsp90 inhibitors-which are barrier enhancers- activate UPR in cells and mice [11,12]. Interestingly, ATF6 delivered global protection against widespread disease [21], and TM protected against brain disorders [22]. Indeed, it was also shown that this compound protects against LPS-induced astrocytic activation and blood–brain barrier hyperpermeability [23]. Recent reviews have summarized the role of UPR activation in lung and cardiovascular disease [10,24,25]. The purpose of the current study is to investigate the effects of TM against LPS-induced endothelial hyper-permeability, based on the hypothesis that TM enhances vascular barrier function. Hence, UPR activation may be of therapeutic value in diseases related to endothelial barrier dysfunction, including ARDS and sepsis.

## 2. Results

### 2.1. TM Enhances Lung Endothelial Integrity

Bovine pulmonary artery endothelial cells were seeded onto gold-plated electrode arrays to form a confluent monolayer. After the cells reach a steady-state resistance, they were treated with either vehicle (0.1% DMSO) or TM (0.5 μM). There was a gradual increase in transendothelial resistance (TEER) values in thoscells that were treated with TM (Figure 1A). The green line indicates the endothelial barrier-enhancing effects of TM in BPAEC.

### 2.2. TM Protects against LPS-Induced Increase in Lung Endothelial Permeability

BPAEC cells were pre-treated with either vehicle (0.1% DMSO) or TM (0.5 μM) for 24 h and post-treated with either vehicle (PBS) or LPS (1 μg/mL). Incubation of the endothelial monolayer with LPS triggers endothelial hyperpermeability (decreased TEER) (Figure 1B). Pre-treatment of BPAECs monolayer with TM (0.5 μM) for 24 h before exposure to LPS (10 μg/mL) suppresses the LPS-induced hyperpermeability.

### 2.3. TM Protects against LPS-Induced Induction of Paracellular Permeability in Lung Endothelial Cells

BPAECs were seeded onto the transwell insert in 24-well culture plates and allowed to form a monolayer on the insert. Then the cells were pre-treated with either vehicle (0.1% DMSO) or TM (1 μM) for 48 h and post-treated with either vehicle (PBS) or LPS (1 μg/mL). BPAEC monolayer was then incubated with FITC-dextran (70 kDa, 1 mg/mL) for 20 min and fluorescein intensity was measured. Cells that were exposed to LPS showed increased paracellular permeability reflected in the higher fluorescence intensity of the media, collected from the receiver well, compared to vehicle treated cells. Pre-treatment of BPAECs with TM (1 μM) for 48 h suppressed the increased paracellular permeability due to LPS treatment (Figure 1C).

### 2.4. TM Suppresses LPS-Induced Phosphorylation of VE-Cadherin 

BPAEC were treated with either vehicle (0.1% DMSO) or TM (1 μM) for 24 h prior to the exposure to either vehicle (PBS) or LPS (1 μg/mL) for 1 h. Treatment of BPAECs with LPS increases the phosphorylation of VE-cadherin (pVE-cad) at tyrosine 685 residue (Y-685) (Figure 1D). However, pre-treatment of the cells with TM (1 μM) for 24 h before LPS (1 μg/mL) treatment protects against the LPS-induced phosphorylation of VE-cadherin. 

### 2.5. TM Counteracts LPS-Induced UPR Suppression

BPAEC were treated with either vehicle (PBS) or LPS (1 μg/mL) for 1 h after the pre-treatment of the cells with TM (1 μM) for 24 h. Western blot analysis from Figure 2A indicates that LPS suppresses UPR in BPAEC as indicated by the downregulation of BiP after 1 h of LPS exposure. Pre-treatment of those cells with TM (1 μM) counteracts this effect and induces BiP levels.

### 2.6. TM Suppresses LPS-Induced Activation of Cofilin in BPAEC

Phosphorylation of cofilin at Ser3 residue deactivates its actin severing activity [26]. BPAEC that were exposed to LPS (1 μg/mL) for 1 h induced the dephosphorylated form of cofilin. Pretreatment of the cells with TM (1 μM) for 24 h counteracted the LPS-induced activation of cofilin. Treatment of BPAEC with TM (1 μM) alone induced phosphorylation (deactivation) of cofilin (Figure 2B). 

### 2.7. LPS-Induced Formation of Actin Stress Fibers Is Counteracted by TM in Lung Endothelial Cells

BPAEC were treated with TM (1 μM) for 24 h prior to a 1 h LPS (1 μg/mL) exposure. Figure 2C demonstrates that LPS induces the activation of MLC2 by phosphorylation. On the other hand, cells that were pre-treated with TM opposed that effect, and TM treatment suppressed the phosphorylation of MLC2 (Figure 2C).

### 2.8. TM Suppresses LPS-Induced Activation of STAT3 in BPAEC

The signal transducer and activator of transcription 3 (STAT3) has been demonstrated to exert crucial roles in inflammation and endothelial cell dysfunction [27]. Cells that were treated with LPS (1 μg/mL) for 1 h showed increased phosphorylation of STAT3 (pSTAT3) (Figure 2D). Pretreatment of those cells with TM (1 μM) for 24 h counteracted the LPS-induced activation of STAT3. TM alone suppressed STAT3 phosphorylation. 

### 2.9. TM Protects against LPS-Induced Activation of Cofilin and MLC2 in Mice

Male C57BL/6 mice were treated with an intratracheal (i.t.) administration of vehicle (saline) or LPS (1.6 mg/kg), and 24 h later the mice received either vehicle (10% DMSO in saline) or TM (0.4 mg/kg) via an intraperitoneal (i.p.) injection. The mice were sacrificed 48 h after LPS treatment and lung tissues were collected. The western blot analysis revealed that LPS increases the activation of cofilin and MLC2 in lung tissues. Post-treatment with TM counteracted the aforementioned LPS-triggered events (Figure 3A,B).

### 2.10. TM Inhibits LPS-Induced Phosphorylation of VE-Cadherin in Mouse Lungs

C57BL/6 mice received either vehicle (saline) or LPS (1.6 mg/kg) via an intratracheal (i.t.) injection, and 24 h later the mice were treated with an intraperitoneal (i.p.) injection of either vehicle (10% DMSO in saline) or TM (0.4 mg/kg, dissolved in 10% DMSO). Figure 3C indicates that mice treated with LPS (1.6 mg/kg) for 24 h showed increased phosphorylation of VE-cadherin (pVE-cad) in the lungs. However, post-treatment with TM counteracted the LPS-induced phosphorylation of VE-cadherin. 

### 2.11. TM Protects against LPS-Induced Activation of STAT3 Signaling in Mouse Lungs

Wild type C57BL/6 mice were subjected to LPS (1.6 mg/kg) treatment via an intratracheal (i.t.) administration for 24 h, followed by post-treatment with either vehicle (10% DMSO in saline) or TM (0.4 mg/kg, dissolved in saline) for 24 h. Our results demonstrate that LPS increases the phosphorylation of STAT3 in the lungs of the mice (Figure 3D). Post-treatment with TM counteracted the inflammatory effects of LPS. Moreover, TM reduced STAT3 activation.

## 3. Discussion

Proteostasis is critical for cell viability, which is endangered due to internal or external stimuli of challenging environmental factors. Brefeldin A, TM, thapsigargin, A23187, AB5 subtilase cytotoxin have been identified to pharmacologically manipulate UPR [2,14,28]. The calcium ionophore A23187 triggers ER stress by mediating the efflux of Ca2+ from the ER lumen, which ultimately induces UPR [29]. UPR inducer TM is isolated from Streptomyces lysosuperificus [30]. This multifaceted mechanism-UPR-has been previously shown to modulate LPS-induced endothelial hyperpermeability [1], and Kifunensine-a UPR suppressor-compromises barrier function [19]. GHRH antagonists utilize UPR to exert their protective activities in the lungs [20]. Hsp90 inhibitors, which are barrier enhancers, have been shown to activate UPR in cells and mice [11,12]. Interestingly, ATF6—a UPR sensor—delivered global protection against widespread disease [21]. TM was shown to protect against brain disorders [22] and LPS-induced astrocytic activation and blood–brain barrier hyperpermeability [23]. Interestingly, TM attenuates LPS-stimulated induction of nitric oxide synthase (iNOS), cyclooxygenase-2 (COX-2), interleukin-1 beta (IL-1β), and TNF-α [31]. Recent reviews have summarized the emerging role of UPR activation in lung and cardiovascular disease [2,10,24,25]. On the other hand, Maciel et al. demonstrated that TM could induce lung fibrosis [32]. The investigators utilized immortalized epithelial cells MLE12, in contrast to our study where we worked with non-immortalized bovine pulmonary artery endothelial cells. Moreover, the endothelial cells are characterized by unique properties [33], which are considered unparalleled to other cell types.

GRP78/BiP marks UPR activation, and its reduction impairs the anti-inflammatory effects of geranylgeranylacetone in experimental models of renal ischemia and colitis [31]. O-GlcNAcylation of VE-Cadherin and BiP cell surface translocation have been linked to ER stress-mediated barrier dysfunction [34]. BiP triggers the protein kinase B (Akt) pathway and reduces oxidative stress in cardiac I/R injury [35]. As BiP is targeted by TM, we sought to determine whether TM affects the LPS-induced downregulation of BiP. Our results demonstrate that LPS significantly suppresses BiP in bold contrast to TM, which opposed that effect (Figure 2A). Moreover, it has been reported that thapsigargin, TM, and SubAB upregulate A20; an endogenous inhibitor of NF-kB [16]. A20-deficient mice experience multi-organ inflammation and are vulnerable to sublethal doses of TNF-α [36].

Endothelial permeability is orchestrated by highly regulated transcellular and paracellular pathways. Transendothelial fluid sieving varies remarkably in different vascular beds. In healthy organs, plasma fluid extravasation is diminished upon the expiration of the stimuli. However, it persists in chronic inflammation, allowing passage to macromolecules larger than 3nm (albumin, IgG). That process is known as transcytosis or vesicular transport [37]. LPS-induced hyperpermeability of albumin was observed via lung microvessels, inducing protein-rich pulmonary edema [38]. Inflammatory mediators disrupt EJs, which in turn weaken the junctional barrier. Our study reports for the first time that TM counteracts LPS-induced lung endothelial barrier hyperpermeability, as reflected in the increased TEER values of the corresponding groups (Figure 1A,B).

To regulate paracellular permeability, AJs integrity is crucial. It limits the passage of solutes smaller than 3 nm. Interestingly, endothelial plasma membrane is almost impermeable to small ionized solutes [37], which are easily interchangeable through fluid-filled clefts. The leukocyte transmigration, fluid transport, and vascular reconstruction are regulated by temporal and spatial expansion of paracellular permeability [39]. Our observations suggest that LPS increases paracellular permeability, as expected. TM prevents that effect (Figure 1C). When VE-cadherin homophilic adhesions are disrupted, the excessive fluid accumulation of the interstitium affects normal respiratory functions [39]. Phosphorylation of VE-cadherin induces actomyosin contraction, which results in actin cytoskeleton remodeling and filamentous actin formation. Those events disassemble AJs [40].

Endothelial barrier integrity is regulated by the small GTPase Ras-related C3 botulinum toxin substrate 1 (Rac1). It induces a peripheral actin cytoskeleton, which in turn enhances the interactions between AJs and TJs. Rac1 stimulates the reannealing of AJs by promoting the reorganization of microfilaments into lamellipodia and filopodia protrusions [41]. Inflammatory mediators upregulate cellular traction inducing mechanical stress [42]. Herein; we observed that LPS attenuates vascular integrity through phosphorylation of VE-cadherin at Y685. TM counteracted that effect in bovine lung cells and in mice (Figure 1D and Figure 3C). It has been reported that the cytoplasmic domain of this adherin junction is disrupted through tyrosine residue 685 (Y685) [43].

By recruiting Rac-specific guanine nucleotide exchange factors (GEFs) VE-cadherin upregulates Rac1 activity, leading to actin filaments polymerization and AJs stabilization [44]. LIM domain kinase is associated with Rac-mediated actin cytoskeletal reorganization by phosphorylating cofilin [26]. Cofilin is an endogenous actin regulatory protein that is crucial for cell protrusion. The severing activity of this 19-kDa ubiquitous protein (cofilin) can generate free actin barbed ends, stimulating actin depolymerization [22]. Knockdown of this cytoskeletal protein by small interfering RNA reduces the release of proinflammatory mediators; including TNFα, iNOS, and cyclooxygenase-2 [23]. Interestingly, post-transcriptional inhibition of iNOS occurred when glial cells were exposed to TM [24]. Cofilin can also stimulate LPS-induced microglial cell activation via NF-κB and janus kinase-signal transducer and activator of transcription (JAK-STAT), producing neurotoxicity. In this study, we showed that TM opposes LPS-induced phosphorylation of STAT3 in BPAEC and mouse lungs (Figure 2D and Figure 3D). In immortalized mouse alveolar epithelial MLE12 cells it was shown that TM could induce lung fibrosis [32].

TM downregulates Ras homolog gene family A (RhoA) [45]. Inflammatory agents (thrombin, LPS) stimulate Rho GTPases and trigger nuclear translocation [46]. Rho-kinase deactivates myosin light chain (MLC) phosphatase, which increases MLC phosphorylation and actin-myosin contractility [45]. Rac1 counteracts RhoA, promoting VE-cadherin trans-interaction, which depends on the junctional localization of p190RhoGAP [19]. VE-cadherin signaling maintains a balance between RhoA and Rac1 at AJs [47]. Moreover, inhibition of UPR potentiates endothelial barrier hyperpermeability and induces the activation of MLC2 and cofilin [1,19,48]. Thrombin-induced phosphorylation of MLC2 precedes the reduction in TEER values [49]. In line with those investigations, we evaluated the effects of TM in the lung endothelium to observe that TM reduces LPS-induced activation of cofilin and phosphorylation of MLC2 both in vitro and in vivo (Figure 2B,C and Figure 3A,B). Previous studies have demonstrated that 1 μg/mL of LPS inflicts cell injury in BPAEC [1,20,50], and 10 μg/mL of LPS induces pulmonary inflammation in mice [11,12,50,51]. Furthermore, the three figures of our paper demonstrate that LPS induced barrier dysfunction in BPAEC (Figure 1B–D and Figure 2B–D) and mouse lungs (Figure 3). Activation of MLC2 (Figure 2C and Figure 3B) and pSTAT3 (Figure 2D and Figure 3D) are associated with inflammatory responses and endothelial barrier disruption [26]. Hence, we conclude that UPR modulation represents a novel therapeutic target against inflammatory lung disease. Further studies in septic and endotoxemic mice will aim to substantiate our findings and propel intense efforts to devise new ways to fight sepsis and ARDS.

## 4. Materials and Methods

### 4.1. Reagents

Tunicamycin (89156-900), anti-mouse IgG HRP-linked whole antibody from sheep (95017-554), anti-rabbit IgG HRP-linked whole antibody from donkey (95017-556), nitrocellulose membranes (10063-173), and RIPA buffer (AAJ63306-AP) were obtained from VWR (Radnor, PA, USA). The phospho-MLC2 (3674S), MLC2 (3672S), phospho-cofilin (3313S), cofilin (3318S), phospho-STAT3 (9145S), STAT3 (4904S), VE-cadherin (2158S) and BiP (3183S) antibodies were purchased from Cell Signaling Technology (Danvers, MA, USA). Lipopolysaccharides (LPS) (L4130) and β-actin antibody (A5441) were purchased from Sigma-Aldrich (St. Louis, MO, USA). Phospho-VE-cadherin (ab119785) from abcam (Waltham, MA, USA).

### 4.2. Cell Culture

Bovine pulmonary artery endothelial cells (BPAEC) were purchased from Genlantis (San Diego, CA, USA). The cells were cultured in DMEM (VWRL0101-0500) supplemented with 10% fetal bovine serum and 1× penicillin/streptomycin. Cultures were maintained at 37 °C in a humidified atmosphere of 5% CO_2_–95% air. All reagents were purchased from VWR (Radnor, PA, USA).

### 4.3. Animals

Seven-week old wild type C57BL/6 (male) mice were purchased from Envigo (Indianapolis, IN, USA). They were maintained in a 12:12-h light/dark cycle, in pathogen free conditions. The temperature (22–24 °C), and humidity (50–60%) were controlled. All experimental procedures were approved by the University of Louisiana Monroe Institutional Animal Care and Use Committee (IACUC).

### 4.4. In Vivo Treatments

The stock solutions of E. Coli LPS (0111:B4) and tunicamycin were prepared in saline and 10% DMSO, respectively. Mice were treated with vehicle (saline) or LPS (1.6 mg/kg) via intra-tracheal instillation. After 24 h of LPS administration, mice received vehicle (10% DMSO in saline) or tunicamycin (0.4 mg/kg each) via an intra-peritoneal injection and were sacrificed by cervical dislocation after 24 h of treatment. All procedures were approved by the committee on Animal Research at University of Louisiana Monroe.

### 4.5. Western Blot Analysis

Proteins were extracted from the cells or tissues using RIPA buffer, and subsequently were separated according to their molecular weight by electrophoresis onto sodium dodecyl sulfate (SDS-PAGE) Tris-HCl gels. A wet transfer technique was used to transfer the proteins onto the nitrocellulose membranes, which were incubated for 1 h at room temperature in a solution of 5% non-fat dry milk, and were consequently exposed overnight to the appropriate primary antibodies (1:1000) at 4 °C. The following day, the membranes were incubated with the corresponding secondary antibodies (1:2000) and were then exposed to SuperSignal^TM^ West Pico PLUS chemiluminescent substrate (PI34578). The signal for the protein bands were detected in a ChemiDoc^TM^ Touch Imaging System from Bio-Rad (Hercules, CA, USA). The β-actin was the loading control unless otherwise stated in the graph of densitometry. All reagents were obtained from VWR (Radnor, PA, USA).

### 4.6. Fluorescein Isothiocyanate (FITC)-Dextran Permeability Assay

Paracellular influx across the BPAEC monolayers was measured using the transwell assay system in 24-well culture plates. A total of 200,000 cells were seeded on each insert, and were allowed to form monolayer for 24 h. The cells were treated with either vehicle (0.1% DMSO) or tunicamycin (1 µM) for 48 h. Consequently, those cells were exposed to either vehicle (PBS) or LPS (1 µg/mL) for 1 h prior to incubation with FITC-dextran (70 kDa, 1 mg/mL) for 20 min. To measure fluorescence, 100 mL of media was collected from each receiver well after the incubation. The intensity of the fluorescence was measured in a Synergy H1 Hybrid Multi-Mode Reader from Biotek (Winooski, VT, USA). The excitation and emission wavelengths were 485 nm and 535 nm, respectively.

### 4.7. Measurement of Endothelial Barrier Function

The transendothelial resistance of the monolayers was estimated by electric cell-substrate impedance sensing (ECIS), utilizing the model ZΘ (Applied Biophysics, Troy, NY, USA). All experiments were conducted on confluent cells that had reached a steady-state resistance of at least 800 Ω.

### 4.8. Densitometry and Statistical Analysis

Image J software (National Institute of Health) was used to perform densitometry of immunoblots. All data is expressed as Means ± SEM (standard error of the mean). Student’s *t*-test was used to determine statistically significant differences among the groups. A value of *p* < 0.05 was considered significant. GraphPad Prism (version 5.01) was used to analyze the data. The letter n represents the number of experimental repeats.

## 5. Conclusions

Therapies that stochastically accelerate the repair of the endothelial microvasculature may deliver new therapeutic possibilities towards ARDS and sepsis. The effects of UPR on the regulation of lung endothelial barrier remain largely unknown, as well as the effects of ATF6, IRE1α and PERK in that context. Our study introduces the protective role of TM in the inflamed lung endothelium, to uncover exciting possibilities in the spectrum of the corresponding pathologies. We demonstrated that TM enhances transcellular and paracellular permeability, opposes LPS-triggered endothelial dysfunction, and counteracts lung inflammation. Future studies will delineate the effects of UPR sensors in the pathophysiology of human disease, as well as will shed light on the output of BiP modulation—the downstream UPR activation target—in barrier function.

## Figures and Tables

**Figure 1 pharmaceuticals-15-00134-f001:**
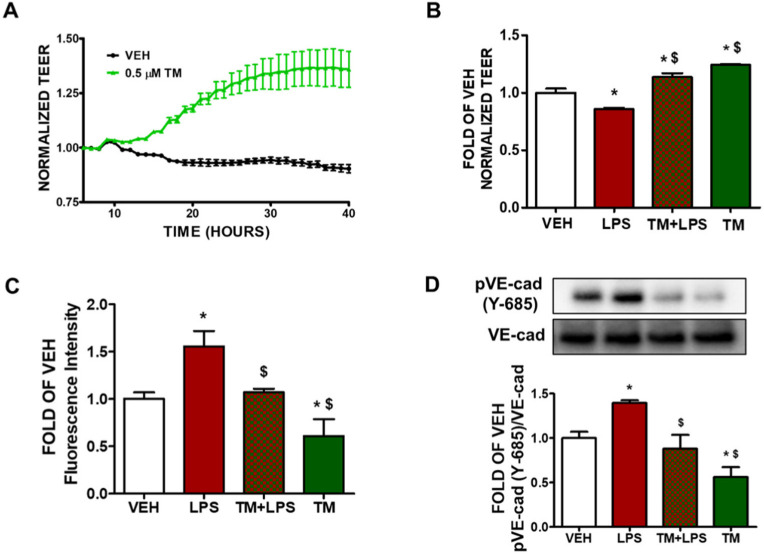
Effects of tunicamycin and LPS on lung endothelial permeability. (**A**) BPAECs were grown on gold-plated ECIS arrays to form a confluent monolayer. Those cells were exposed to either vehicle (VEH) (0.1% DMSO) or tunicamycin (TM) (0.5 μM). A gradual decrease in BPAEC permeability (increased TEER) was observed in the TM-treated cells. *n* = 3 per group; means ± SEM. (**B**) BPAECs were treated with either vehicle (0.1% DMSO) or TM (0.5 μM) for 24 h prior to treatment with either vehicle (PBS) or LPS (10 μg/mL). LPS exposure decreased TEER values (increased permeability), while TM pretreatment prevented LPS-triggered barrier dysfunction as reflected in the higher TEER values. * *p* < 0.05 vs. vehicle (VEH) and ^$^
*p* < 0.05 vs. LPS. *n* = 3 per group; means ± SEM. (**C**) BPAECs were seeded onto transwell inserts of a 24-well culture plate. After 24 h, the cells were treated with either vehicle (VEH) (0.1% DMSO) or TM (1 µM) for 48 h. LPS (1 µg/mL) was added in the media for 1 h; followed by the addition of 70 kDa FITC-dextran (1 mg/mL). 20 min after FITC-dextran addition, 100 mL of basal media was removed and the fluorescence intensity was measured. * *p* < 0.05 vs. vehicle (VEH) and ^$^
*p* < 0.05 vs. LPS. Means ± SEM, *n* = 3. (**D**) Western blot analysis of phosphorylated VE-cadherin (pVE-cad) and VE-cadherin (VE-cad) in BPAEC treated with either vehicle (0.1% DMSO) or TM (1 μΜ) prior to a 1 h exposure to either vehicle (PBS) or LPS (1 μg/mL). The blots shown are representative of three independent experiments. The signal intensity of the protein bands was analyzed by densitometry. Protein levels of pVE-cad were normalized to total VE-cad. * *p* < 0.05 vs. vehicle (VEH) and ^$^
*p* < 0.05 vs. LPS. Means ± SEM.

**Figure 2 pharmaceuticals-15-00134-f002:**
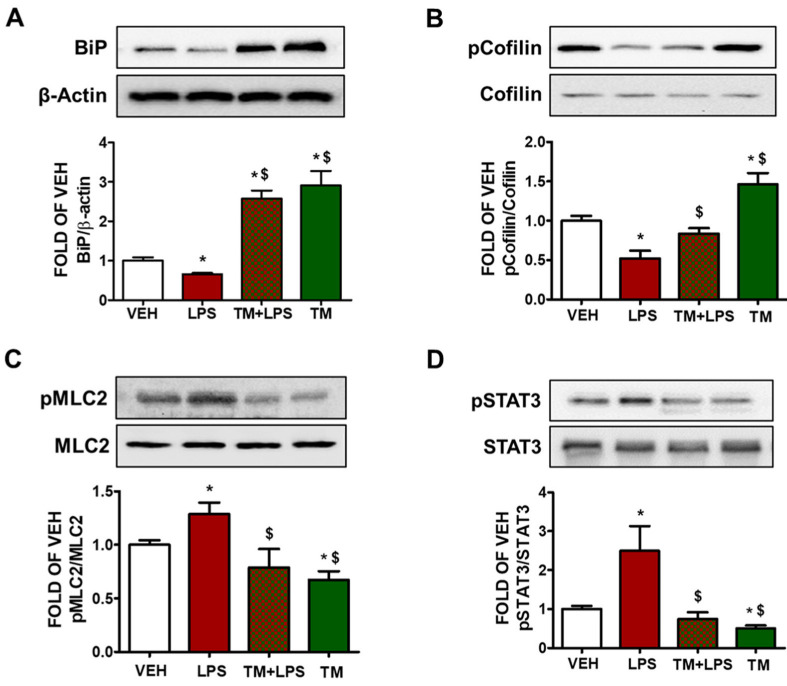
Effects of tunicamycin in LPS-induced lung endothelial inflammation. (**A**) Western Blot analysis of BiP and β-actin in BPAEC treated with either vehicle (0.1% DMSO) or tunicamycin (TM) (1 μΜ) prior to treatment with either vehicle (PBS) or LPS (1 μg/mL) for 1 h. The blots shown are representative of four independent experiments. The signal intensity of the protein bands was analyzed by densitometry. Protein levels of BiP were normalized to β-actin. * *p* < 0.05 vs. vehicle (VEH) and ^$^
*p* < 0.05 vs. LPS. Means ± SEM. Western Blot analysis of (**B**) phosphorylated cofilin (pCofilin) and cofilin, (**C**) phosphorylated MLC2 (pMLC2) and MLC2, (**D**) phosphorylated STAT3 and STAT3. BPAEC were pre-treated with either vehicle (0.1% DMSO) or TM (1 μΜ) for 24 h and post-treated with either vehicle (PBS) or LPS (1 μg/mL) for 1 h. The blots shown are representative of three independent experiments. The signal intensity of the protein bands was analyzed by densitometry. Protein levels of p-Cofilin, pMLC2, and pSTAT3 were normalized to cofilin, MLC2 and STAT3, respectively. * *p* < 0.05 vs. vehicle (VEH) and ^$^
*p* < 0.05 vs. LPS. Means ± SEM.

**Figure 3 pharmaceuticals-15-00134-f003:**
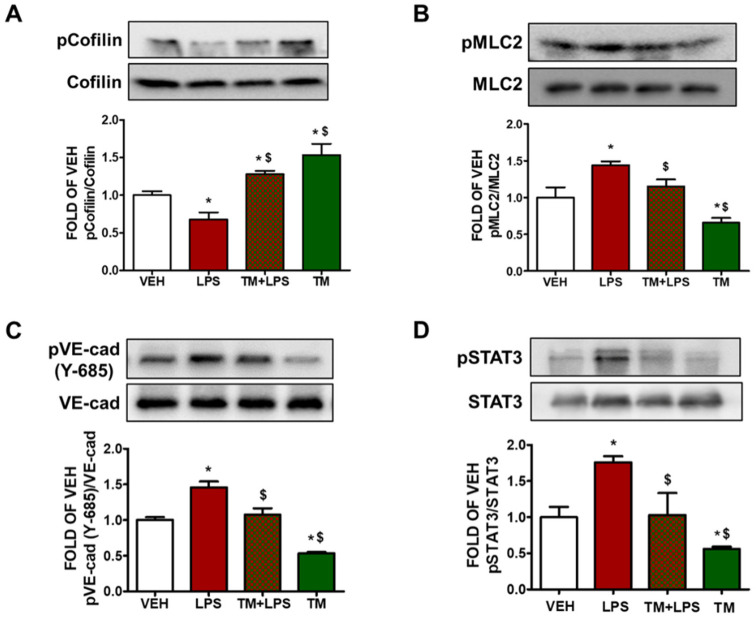
Effects of tunicamycin in LPS-induced inflammation in mouse lungs. Western Blot analysis of (**A**) phosphorylated cofilin (pCofilin) and cofilin, (**B**) phosphorylated MLC2 (pMLC2) and MLC2, (**C**) phosphorylated VE-cadherin (pVE-cad) and VE-cadherin (VE-cad), (**D**) phosphorylated STAT3 (pSTAT3) and STAT3 expression in lungs retrieved from mice 48 h after an intratracheal injection of either vehicle (saline) or LPS (1.6 mg/kg); and post-treated (24 h after LPS) with an intraperitoneal injection of either vehicle (10% DMSO in saline) or tunicamycin (TM) (0.4 mg/kg each, dissolved in 10% DMSO). The signal intensity of the protein bands was analyzed by densitometry. Protein levels of pCofilin, pMLC2, pVE-cad and pSTAT3 were normalized to cofilin, MLC2, VE-cad and STAT3, respectively. * *p* < 0.05 vs. vehicle (VEH) and ^$^ *p* < 0.05 vs. LPS, *n* = 3 mice per group. Means ± SEM.

## Data Availability

Data is contained within the article.

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
