# Peer review of "Tunicamycin Protects against LPS-Induced Lung Injury"

_pharmaceuticals, 2022, doi:10.3390/ph15020134_

Round 1

Reviewer 1 Report

Dr Kubra and coworkers explore the effects of Tunicamycin on acute lung injury. however, there are several questions to clarify.

  1. why did the authors want to prove Tunicamycin, there are no enough evidence to explain the pharmacological mechanism in background, and please give more papers to prove the use of Tunicamycin.
  2. the authors used two model, one was based on cell and another was on mice, however, there are no pathologic findings in cell and mice studies
  3. According to other study( Aging (Albany NY). 2018 Aug 27;10(8):2098-2112.),  Tunicamycin could induced lung fibrosis, and how did the authors evaluate the therapeutic effects?
  4. this study did not show the basic results of endothelial permeability and lung weight, et al. 

Author Response

Comment 1: Why did the authors want to prove Tunicamycin, there are no enough evidence to explain the pharmacological mechanism in background, and please give more papers to prove the use of Tunicamycin.

Reply 1: Thank you very much for the suggestion, we have incorporated the folloing information in our manuscript: Tunicamycin induces the activation of the unfolded protein response, which has been previously shown to modulate LPS-induced endothelial hyperpermeability (Ref. Kubra et al., Am J Physiol Cell Physiol. 2021;321(2):C214-C220. doi:10.1152/ajpcell.00142.2021). Moreover, Kifunensine-a UPR suppressor-compromises barrier function (Ref. Akhter et al,. Microvascular research, 132 (2020): 104051. doi:10.1016/j.mvr.2020.104051), GHRH antagonists utilize UPR to exert their protective activities in the lungs, and Hsp90 inhibitors-which are barrier enhancers-activate UPR in cells and mice (Ref. Kubra et al., Cell Signal. 2020;67:109500. doi:10.1016/j.cellsig.2019.109500; Uddin et al., Med Drug Discov. 2020;6:100046. doi:10.1016/j.medidd.2020.100046). Interestingly, ATF6 delivered global protection against widespread disease (Blackwood et al., Pharmacologic ATF6 activation confers global protection in widespread disease models by reprograming cellular proteostasis. Nat Commun. 2019 Jan 14;10(1):187. doi: 10.1038/s41467-018-08129-2). Furthermore, TM was shown to protect against brain disorders (Ref. Hang et al., Autophagy. 2014 Oct 1;10(10):1801-13. doi: 10.4161/auto.32136). Indeed, it was also shown that this compound protects against LPS-induced astrocytic activation and blood-brain barrier hyperpermeability (Ref. Wang et al., Front Cell Neurosci. 2018 Jul 27;12:222. doi: 10.3389/fncel.2018.00222). Recent reviews have summarized the role of UPR activation in lung and cardiovascular disease (Ref. Kubra et al., Cell Signal. 2020;73:109699. doi:10.1016/j.cellsig.2020.109699; Barabutis N. Front Med (Lausanne). 2020;7:344. doi:10.3389/fmed.2020.00344; Akhter et al., Curr Res Cell Biol. 2020;1:100003. doi:10.1016/j.crcbio.2020.100003).

Comment 2: The authors used two model, one was based on cell and another was on mice, however, there are no pathologic findings in cell and mice studies.

Reply 2: Thank you very much for the suggestion, we have incorporated the following information in our manuscript: We have previously shown that this particular dosage of LPS induces cell inflammation and lung injury in mice (Ref. Barabutis et al., Cytokine. 2019;113:427-432. doi:10.1016/j.cyto.2018.10.020; Rafikov et al., J Biol Chem. 2014;289(8):4710-4722. doi:10.1074/jbc.M114.547596; Akhter et al., Pharmaceuticals (Basel). 2021;14(6):522. oi:10.3390/ph14060522). Furthermore, the activation of MLC2 (Figure 2C and 3B) and pSTAT3 (Figure 2D and 3D) are associated with inflammatory responses and endothelial barrier disruption (Barabutis et al., Am J Physiol Lung Cell Mol Physiol. 2016 Nov 1;311(5):L832-L845. doi: 10.1152/ajplung.00233.2016).

Comment 3: According to other study Aging (Albany NY). 2018 Aug 27;10(8):2098-2112., Tunicamycin could induced lung fibrosis, and how did the authors evaluate the therapeutic effects?

Reply 3: Our apologies for the omission of the important paper by Maciel et al, entitled “Impaired autophagic activity and ATG4B deficiency are associated with increased endoplasmic reticulum stress-induced lung injury”. Aging (Albany NY). 2018 Aug 27;10(8):2098-2112. doi: 10.18632/aging.101532. That paper is now cited in the discussion of our paper. In that paper the authors utilized the immortalized epithelial cells MLE12, while we worked with non-immortalized bovine pulmonary artery endothelial cells. Hence, our results correspond to a completely different context. Furthermore, the endothelial cells are characterized by unique properties (Ref. McCarron et al., Trends Pharmacol Sci. 2017 Apr;38(4):322-338. doi: 10.1016/j.tips.2017.01.008.) which are unparalleled to other cell types.

That information is now included in our manuscript.

Comment 4: This study did not show the basic results of endothelial permeability and lung weight, et al.

Reply 4: It was previously shown that LPS induces endothelial permeability and lung injury in mice (Ref. Barabutis et al., Cytokine. 2019;113:427-432. doi:10.1016/j.cyto.2018.10.020; Rafikov et al., J Biol Chem. 2014;289(8):4710-4722. doi:10.1074/jbc.M114.547596; Akhter et al., Pharmaceuticals (Basel). 2021;14(6):522. doi:10.3390/ph14060522). In the current study, we have shown that LPS induces endothelial hyperpermeability that is counteracted by TM as reflected in increased TEER values (Figure 1B).

Reviewer 2 Report

I have carefully evaluated the paper on the role of tunicamycin in endothelial barrier dysfunction, written by Kubra et al.

In this paper, the authors use bovine endothelial cells and mice subjected to tunicamycin and LPS, to evaluate the role of tunicamycin in LPS-triggered endothelial hyperpermeability and inflammation.

The authors demonstrate that UPR induction due to tunicamycin opposes the LPS-induced cofilin activation, suppresses the LPS-induced formation of the filamentous actin, as well as the activation of STAT3, both in vivo and in vitro. Also, they measure paracellular and transendothelial permeability to conclude that TM counteracts the LPS-induced endothelial breakdown.

The findings are of high quality, the interpretations of the results provided by the authors correspond to the findings, the discussion is complete, and the references are up to date. It appears that the authors are experienced on such studies, their results are novel and hold the potential to lead to novel possibilities in the development of novel approaches to treat diseases related to endothelial barrier dysfunction (i.e., sepsis, lung injury) and lung inflammatory disease.

Author Response

Thank you.

Reviewer 3 Report

The article has an  huge impact on medical field

But some errors appear, that must be corrected:

  1. please specify during abstract the meaning for TM
  2. please specify more clear the results of the study in the abstract-starting with line 18
  3. line 70-specify the full name for TNF-alpha and NF-kB
  4. LPS is once abbreviated at line 42, so it is not necessary at line 75
  5. at the end of introduction please specify more clearly the aim of the study. 
  6. at line 186 is mentioned an wild type of mice, but this aspect  is not mentioned at materials and methods-it should be
  7. line 200- TNF alpha appears with full name, it is not necessary
  8. line 206- full name for Akt
  9. lines  251  and 252 appears full name for TNF-alpha and iNOS-correct
  10. line 254-full name for JAK
  11. it was difficult for me to read the article, having put the results after the introduction and then materials and methods
    why weren't they written immediately after the introduction?
    the following order was not better: introduction, materials and methods, results, discussions
  12. Please  a few more details about the results  at the conclusions

Author Response

The article has an huge impact on medical field

But some errors appear, that must be corrected:

Comment 1: please specify during abstract the meaning for TM

Reply 1: The abstract was updated to address your kind comment.

Comment 2: please specify more clear the results of the study in the abstract-starting with line 18

Reply 2: We have included the findings of our study in response to your comment.

Comment 3: line 70-specify the full name for TNF-alpha and NF-kB

Reply 3: The requested information is now included in 5th paragraph of the introduction.

Comment 4: LPS is once abbreviated at line 42, so it is not necessary at line 75

Reply 4: We updated the “introduction” section.

Comment 5: at the end of introduction please specify more clearly the aim of the study. 

Reply 5: In response to your comment, we explained our hypothesis in the 6th paragraph            of the introduction

Comment 6 :at line 186 is mentioned an wild type of mice, but this aspect  is not mentioned at materials and methods-it should be

Reply 6: Please see the updated paragraph of “materials and methods” section.

Comment 7: line 200- TNF alpha appears with full name, it is not necessary

Reply 7: We updated the “Discussion” to address your comment

Comment 8: line 206- full name for Akt

Reply 8: We added the full name of Akt.

Comment 9: lines  251  and 252 appears full name for TNF-alpha and iNOS-correct line 254-full name for JAK

Reply 9: The requested information is now included in those lines.

Comment 10: it was difficult for me to read the article, having put the results after the introduction and then materials and methods
why weren't they written immediately after the introduction?
the following order was not better: introduction, materials and methods, results, discussions

Reply 10: We apologize for that, but we would like to follow journal’s instructions.

Comment 11: Please  add  few more details about the results  at the conclusions

Reply 11: In the conclusions, we described our results to follow the referee’s comments.

Round 2

Reviewer 1 Report

it's better than before

Reviewer 3 Report

The authors answered all my comments